# Flow Matching for Probabilistic Monocular 3D Human Pose Estimation

**Cuong Le**                                                                    *cuong.le@liu.se*
*Department of Electrical Engineering*
*Linköping University*

**Pavlo Melnyk**                                                             *pavlo.melnyk@liu.se*
*Department of Electrical Engineering*
*Linköping University*

**Bastian Wandt**                                                      *bastianwandt@gmail.com*
*Independent researcher*

**Mårten Wadenbäck**                                                  *marten.wadenback@liu.se*
*Department of Electrical Engineering*
*Linköping University*

**Reviewed on OpenReview:** *https://openreview.net/forum?id=UlpH4XBLR4*

## Abstract

Recovering 3D human poses from a monocular camera view is a highly ill-posed problem due to the depth ambiguity. Earlier studies on 3D human pose lifting from 2D often contain incorrect-yet-overconfident 3D estimations. To mitigate the problem, emerging probabilistic approaches treat the 3D estimations as a distribution, taking into account the uncertainty measurement of the poses. Falling in a similar category, we proposed FMPose, a probabilistic 3D human pose estimation method based on the flow matching generative approach. Conditioned on the 2D cues, the flow matching scheme learns the optimal transport from a simple source distribution to the plausible 3D human pose distribution via continuous normalizing flows. The 2D lifting condition is modeled via graph convolutional networks, leveraging the learnable connections between human body joints as the graph structure for feature aggregation. While trade-offs between processing time and precision exist, already in the equal-accuracy comparison, FMPose exhibits significantly faster processing time than the diffusion model, and also offers another faster and more accurate configuration. Experimental results show major improvements of our FMPose over current state-of-the-art methods on two common benchmarks for 3D human pose estimation, namely Human3.6M, MPI-INF-3DHP. Additionally, FMPose shows competitive performance on the more challenging 3DPW dataset. The code implementation is available at this GitHub repository.

## 1 Introduction

In computer vision, 3D human pose estimation (HPE) is a long-standing topic that plays a crucial role in a wide range of downstream tasks, including autonomous driving, robotics and public safety (Wang et al., 2024; Andersson et al., 2021). Recovering 3D human poses from monocular camera view is an ill-posed problem because of one major challenge: *the depth uncertainty* along the optical axis. Prior studies, many of which originate from Martinez et al. (2017), address the issue by leveraging machine learning models to estimate the most probable 3D human pose, given either images or estimated 2D poses as input. However, due to the ill-posed nature of the task, predicting a single 3D pose would lead to incorrect and overly confident estimations, especially when encountering occlusions or highly ambiguous scenarios. To address

the problem, emerging probabilistic 3D HPE approaches instead treat the set of 3D pose estimations as a distribution, essentially capturing all possible poses with their corresponding uncertainty measurements. Probabilistic 3D HPE can contribute greatly to downstream tasks, by reinforcing the downstream models with the awareness of a wider range of possible poses, beyond just the maximum likelihood estimate, to make better decisions (Bramlage et al., 2023; Gu et al., 2024).

Recent probabilistic approaches estimate the 3D pose distribution via generative approaches such as conditional autoencoder (Sharma et al., 2019), normalizing flows (Wehrbein et al., 2021), and diffusion models (Ci et al., 2023; Holmquist & Wandt, 2023). The generative models are conditioned on 2D poses estimated from the input images to address the 2D-3D lifting task. The distribution of the plausible 3D poses is often approximated via multi-hypothesis estimations, practically drawing a set of 3D pose hypotheses for a given 2D cue. From the set of hypotheses, the probabilistic methods essentially encapsulate the uncertainty of the true plausible 3D pose, leading to more reliable estimations for downstream use. Despite the advantages, generative models are generally difficult to train and deploy properly, often requiring complex architectures, expensive stochastic processes, or sub-optimal generation quality. Flow matching, a recent breakthrough by Lipman et al. (2023), which has the potential to greatly benefit the task of probabilistic 3D HPE using continuous normalizing flows with optimal transport, has not been explored in-depth.

Compared to diffusion models, which solves a complicated stochastic differential equation (SDE), the continuous normalizing flows with optimal transport provide a more stable and accurate framework for learning the mapping between distributions via a simple ordinary differential equation (ODE) (Chen et al., 2018) along a straight optimal path (Lipman et al., 2023). To this end, we propose a generative approach, namely FMPose, based on the continuous normalizing flows trained by flow matching with optimal transport (OT) for the probabilistic monocular 3D HPE. Specifically, our flow model is used to generate high-quality 3D human pose hypotheses, conditioned on the cues extracted from the full probabilistic estimation of 2D poses.

Inspired by prior work by Holmquist & Wandt (2023), our model is supported by 2D pose observations in the form of probability heatmaps, as opposed to earlier approaches that only consider the maximal arguments (Martinez et al., 2017), or Gaussian fitting (Wehrbein et al., 2021). We create the lifting conditional vector using graph convolutional networks (GCN) (Kipf & Welling, 2017), using the extracted top-$k$ arguments from the 2D heatmaps as inputs. The GCN extract features through a learnable connected graph, with the nodes representing human joints. The graph connections, which can also be understood as relations between the joints, are learned from zero initialization.

We summarize our contributions as follows: 1) We are the first to use the stable and accurate continuous flow model with optimal transport for probabilistic 3D HPE, to the best of our knowledge; 2) We propose a 2D-3D lifting condition formulated via graph convolution networks, effectively integrating the learnable connections between human joints for accurate 3D pose hypothesis generation; and 3) Our FMPose sets new state-of-the-art performance for single-frame multi-hypothesis 3D HPE across multiple popular benchmarks.

## 2 Related work

Monocular 3D human pose estimation consists of two main directions: 1) direct estimation from input images (Mehta et al., 2017b; Rogez et al., 2019) or 2) 2D-3D lifting (Martinez et al., 2017). The proposed FMPose belongs to the category of 2D-3D lifting.

### 2.1 2D-3D pose via lifting

Compared to 3D HPE, 2D HPE is a much more mature research area due to the large amount of available 2D annotations compared to 3D poses. Pre-extracting the 2D pose also help the lifting model to solely focus on estimating the 3D pose, limiting the influence of the surroundings. There exists a subfield of 2D-3D lifting that leverages temporal information to estimate the 3D pose. All of the methods in this category use a sequence of frames as input to predict the middle frame 3D pose, using temporal convolutions (Pavllo et al., 2019; Yeh et al., 2019; Liu et al., 2020), or transformers (Zheng et al., 2021; Zhang et al., 2022; Zhao et al., 2023; Gong et al., 2023). Despite their good performance, these methods are limited in real-world usages due to the costly processing of video input.

Single-image approaches are more suitable for downstream tasks because of the real-time configuration, especially for image data captured in-the-wild. Single-image 2D-3D pose lifting often involves a neural network model to predict the most likely 3D correspondence given the estimated 2D pose. Prior work from Martinez et al. (2017) lays a baseline for the field by using a simple 2-layer ResNet architecture to predict a 3D pose from a single 2D input pose. Chen & Ramanan (2017) finds a closely matching 3D solution given a 2D input and a library of 3D poses. Some recent methods (Cai et al., 2019; Xu et al., 2020) utilize the natural connections of human skeletons via GCN to improve the estimations. A limitation of the mentioned approaches is their prediction of a single, most likely 3D pose, which can be over-confidently incorrect for challenging scenarios. In contrast, our proposed method estimates a distribution of possible 3D poses, fully capturing the uncertainty of the predictions.

## 2.2 Probabilistic 3D pose estimation

Unlike deterministic single-image approaches, probabilistic 3D HPE leverages the heatmap-based 2D detectors (Sun et al., 2019) to predict the corresponding 3D pose distribution (Simo-Serra et al., 2012; Jahangiri & Yuille, 2017). Recent work addresses the probabilistic estimation via generative machine learning that produces multiple hypotheses to approximate the plausible 3D pose distribution. Li & Lee (2019) use a multimodal mixture density network to learn the posterior distribution, basing their hypotheses on a set of mixing coefficients and parameters of the Gaussian kernels. Sharma et al. (2019) employs a variational autoencoder to generate 3D hypotheses conditioned on 2D poses and ranking the 3D poses based on the ordinal relation between joints in the input image. Kolotouros et al. (2021) reconstruct the 3D pose from a volumetric human model SMPL (Loper et al., 2015), using a conditional normalizing flow architecture given input images. Similarly, Wehrbein et al. (2021) also use the conditional normalizing flow to estimate the plausible 3D pose distribution. The conditioning for the flow is realized via fitted 2D Gaussians on the 2D detector's heatmaps. The 2D Gaussians are used to identify highly ambiguous poses based on the variances.

Diffusion models are also a popular approach to generating 3D hypotheses. Ci et al. (2023) introduces GFPose that learns to de-noise from a source Gaussian distribution to plausible 3D poses through the Denoising Score Matching (Vincent, 2011). The method requires very expensive and fragile solvers for the reverse-time SDE. In a similar fashion, DiffPose, proposed by Holmquist & Wandt (2023), learns to denoise by predicting the scheduled added noise, following the Denoising Diffusion Probabilistic Modeling (DDPM) approach (Ho et al., 2020). While using the noise scheduler from DDPM helps reduce the computational cost significantly, DiffPose still solves a complicated stochastic denoising process that causes sub-optimal generative quality. Furthermore, the stability in the DDPM used in DiffPose heavily depends on the quality of the noise scheduling process, and requires a high number of denoising steps to generate 3D pose hypotheses. In contrast, we propose modeling the mapping from the source Gaussian distribution to the plausible 3D pose distribution with continuous normalizing flows learned via optimal transport trajectories using the flow matching framework (Lipman et al., 2023). Based on the experimental results from this study, the straight OT probability path of FMPose demonstrates the better performance, in both prediction accuracy and model complexity, for the task of 2D-3D keypoint lifting compared to the stochastic probability path of DiffPose.

## 3 Method

We aim to learn the mapping, i.e. the flow, from an easy-to-sample source distribution to the posterior distribution of plausible 3D human poses. To realize the 2D-3D lifting, we first apply an off-the-shelf 2D joint estimator HRNet (Sun et al., 2019) to generate a set of 2D heatmaps, representing the probability of each joint location, and subsequently use this information to learn the corresponding 3D poses. The 2D-3D lifting condition is extracted from the heatmaps via GCNs, leveraging the learnable human joint connectivity for feature aggregation. In the common generative settings (Wehrbein et al., 2021; Holmquist & Wandt, 2023), the source distribution is often selected to be the Gaussian distribution. The optimal transport from the Gaussian distribution to the 3D pose distribution is a straight trajectory, demonstrated via the linear interpolation between the two distributions. The neural network model is designed to learn the optimal transport (OT) moving direction given only the current pose states and the corresponding lifting condition. An overview of FMPose is shown in Figure 1.

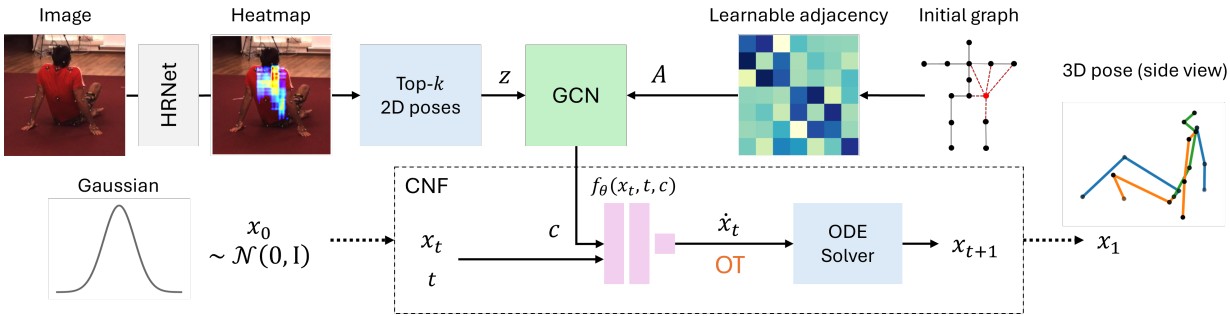

Figure 1: The overview of FMPose. From a set of heatmaps predicted by the 2D pose detector HRNet, top-$k$ arguments are used as the input $z$ for the GCN. The GCN then computes the lifting condition vector $c$ by aggregating 2D information via the learned adjacency matrix $A$. Given the condition, the continuous flow model iteratively moves the initial pose $x_0$, drawn from the Gaussian distribution, towards the plausible 3D pose $x_1$ using a numerical ODE solver. The movement direction, velocity $\dot{x}_t$, is estimated by the neural network $f_\theta$ trained with the OT path, given the current state $(x_t, t)$ and the condition $c$ as inputs. Each initial $x_0$ produces a corresponding 3D pose $x_1$, thus we generate hypotheses by sampling $x_0 \sim \mathcal{N}(0, I)$.

### 3.1  Flow Matching

Considering a mapping bounded in $t \in [0, 1]$ from a random variable $x_0 \in \mathbb{R}^{J \times 3}$ to a corresponding 3D human pose $x_1 \in \mathbb{R}^{J \times 3}$ with $J$ being the number of joints. The mapping trajectory from $x_0$ to $x_1$ is a time-dependent diffeomorphic flow, $\phi : [0, 1] \times \mathbb{R}^{J \times 3} \to \mathbb{R}^{J \times 3}$, defined via a neural ODE of the form

$$\frac{d}{dt}\phi(x_0) = \frac{d}{dt}x_t = \dot{x}_t = f_\theta(x_t, t) \, , \tag{1}$$

where $f_\theta$ is a parameterized neural network (NN) that approximates the flow velocity $\dot{x}_t$. Here, $\phi$ is a continuous normalizing flow (CNF) (Chen et al., 2018), mapping a source density to a complex one via push-forward function.

Traditional CNF approaches learn the untractable $\phi$ through the maximum likelihood of $x_1$, which requires an expensive ODE solving process to identify the suitable trajectory of $\phi$, often resulting in highly unstable training. A recent method from Lipman et al. (2023) leverages the conditional flow matching between $x_0$ and $x_1$ via OT to effectively learn the flow velocity without the need for expensive ODE solvers. Naturally, the OT between $x_0$ and $x_1$ is simply the linear interpolation: $x_t = (1 - t)x_0 + tx_1$. The OT flow velocity $u_t$ can be defined as the first-order derivative of the OT path with respect to time, i.e.

$$u_t = \frac{d}{dt}x_t = \frac{d}{dt}((1 - t)x_0 + tx_1) = x_1 - x_0 \, . \tag{2}$$

The learning of the generative flows reduces to a simple regression task based on Equations (1) and (2). Similar to Lipman et al. (2023), the objective function $\mathcal{L}$ is the mean squared errors (MSE) between the output of the NN model $f_\theta(x_t, t)$ and the OT velocity $u_t$, averaging across all human joints:

$$\mathcal{L} = \frac{1}{3J} \sum_{j=1}^{3J} (f_\theta(x_t^j, t, c) - u_t^j)^2 \, , \quad \forall \, 0 \leq t \leq 1 \, , \tag{3}$$

where $c$ is the conditional lifting vector extracted from the heatmaps via GCN. During training, without expensive sequential ODE solving, we can simply draw $t \sim \mathcal{U}(0, 1)$ and use the linear interpolation to compute the state $x_t$ from $x_0 \sim \mathcal{N}(0, I)$ and the ground truth 3D pose $x_1$.

### 3.2  2D-3D lifting condition

The task-specific condition $c$ is modeled via GCN, aggregating the 2D features through a graph, parameterized by an adjacency matrix $A \in \mathbb{R}^{J \times J}$. The entries of $A$ represent the relationship between human joints

towards the lifting task, with higher values denoting more correlation. We first extract a set of $J$ joint-wise heatmap predictions using a 2D detector (Sun et al., 2019). Instead of taking the maximum argument of each heatmap as inputs, we construct an input tensor $z \in \mathbb{R}^{J \times 2k}$ by selecting a set of top $k$ positions based on their probabilities, essentially extracting the most informative arguments of each heatmap. We then project $z$ to a latent embedding $h \in \mathbb{R}^{J \times d}$ using a linear layer. The 2D-3D lifting condition $c \in \mathbb{R}^{d'}$ modeled via GCN is computed as

$$c = \text{Linear} \left( \text{Flatten} \left( \sigma(A \, h \, W) \right) \right) , \tag{4}$$

where $W \in \mathbb{R}^{d \times d}$ is the GCN learning weights, and $\sigma$ is the non-linear activation function. The extracted features are then flattened and linearly mapped to the condition vector $c$. The entries of the adjacency matrix $A$ are initialized to zeros and adaptively learn the joint relation together with the full training pipeline. An example of a learned adjacency matrix can be seen in Figure 1. While it is a common practice to leverage the natural human skeleton for the adjacency matrix, i.e. in activity recognition (Chen et al., 2021), through experiments (Table 5) we, however, find the skeleton constraint to be too strong, which limits the performance of FMPose. Please see Appendix B for the visualizations of learn-from-scratch adjacency.

The lifting condition is vital to conduct the 2D-3D lifting task; an ablation study can be found in Table 5. With the condition vector, the ODE of FMPose now becomes

$$\frac{d}{dt} x_t = \dot{x}_t = f_\theta(x_t, t, c) , \tag{5}$$

where the function $f_\theta$ is parameterized with neural networks using inputs $x_t, t, c$ concatenated to one vector.

### 3.3 Solving the continuous flow

At inference, the flow from $x_0$ to $x_1$ becomes a CNF, where the ODE is defined in Equation (5). Because $f_\theta$ is parameterized via a neural network, an analytical integration is not available, and a numerical ODE solver is needed to obtain the final solution of 3D human pose $x_1$. The most well-known ODE solver is Runge–Kutta (RK) due to its fixed time-step integration and simplicity. There exist RK solvers of different orders, but here we demonstrate the stable second-order variant RK2 in the following equation:

$$x_{t+\Delta t} = x_t + f_\theta \left( x_t + \frac{\Delta t}{2} f_\theta(x_t, t), \ t + \frac{\Delta t}{2} \right) \Delta t , \tag{6}$$

where $\Delta t$ is the discretized time step between two intermediate solutions. Iteratively solving for $x_1$ from $x_0$, using Equation (6) with a small time step $\Delta t$, results in a smooth OT mapping trajectory. The trade-off between different ODE solvers is demonstrated via the ablation studies in Table 6.

### 3.4 Implementation details

**2D pose estimation** To maintain comparability with previous methods (Wehrbein et al., 2021; Holmquist & Wandt, 2023), we use the HRNet (Sun et al., 2019) for 2D pose detection. While HRNet is originally trained on MPII (Andriluka et al., 2014), we also leverage the fine-tuned version on Human3.6M from (Wehrbein et al., 2021) to work with this dataset, following the related methods. The output of HRNet is a set of heatmaps corresponding to body joints. Instead of taking only the maximum argument, we order all arguments based on their confidence values and extract the top-$k$ positions. By doing so, we cover the most informative areas of the heatmaps, resulting in an input tensor of shape $\mathcal{R}^{J \times 2k}$ for the GCN module. To adopt the random sampling strategy from DiffPose, we draw a set of $k$ candidates based on the confident scores from the estimated heatmaps of HRNet, outputting a tensor of same shape $\mathcal{R}^{J \times 2k}$. The value $k = 48$ is selected based on the ablation in the Appendix A. The dimension of the condition vector $c$ is 64. The dropout layer is applied after the first linear layer in the GCN module, with the drop rate of 0.01. Top-k or random sampling is used after the HRNet's prediction, on the predicted heatmaps.

**Preprocessing** Following the common protocol (Martinez et al., 2017; Wehrbein et al., 2021; Holmquist & Wandt, 2023), the 2D detection is standardized to zero mean and unit variance. The ground-truth 3D poses are presented in meters in camera coordinates, and are mean-centered individually. The GCN adjacency matrix is initialized with zero entries and learned adaptively during training.

**Training** To keep fair comparisons to previous work, we design the $f_\theta$ similarly to the denoising network in DiffPose (Holmquist & Wandt, 2023), which is a 2-layer ResNet with the hidden dimension of 1024, inspired from Martinez et al. (2017). On Human3.6M, the models are jointly trained for 100 epochs with the AdamW (Loshchilov & Hutter, 2019) optimizer, batch size of 64, and an initial learning rate of $1 \times 10^{-4}$ scheduled to reduce with a factor of 0.1 in the last 10 epochs (determined by the training curve). The activation function is the Swish (a.k.a SiLU) (Elfwing et al., 2018; Ramachandran et al., 2018). On 3DPW, the number of epochs remains 100, while the batch size is 128 and the learning rate is changed to $1 \times 10^{-5}$ scheduled with a reduction factor 0.1 at epoch 80. Training is end-to-end with two sequential modules GCN and ODE Solver. On the NVIDIA A40, the training one hour and consumes 86MB of memory per iteration.

## 4 Experiments

### 4.1 Datasets

Following related studies, we perform benchmarking on the Human 3.6M (Ionescu et al., 2014) and the MPI-INF-3DHP (Mehta et al., 2017a) datasets. We additionally evaluate FMPose on the 3DPW dataset (Marcard et al., 2018), which contains challenging in-the-wild images of human poses to verify the robustness of the proposed method. For a fair comparison to related works (Wehrbein et al., 2021; Holmquist & Wandt, 2023) on Human3.6M, we consider two experimental setups: 1) the common evaluation on every $64^{\text{th}}$ frame of the testing set, and 2) evaluation on the highly ambiguous poses collected by (Wehrbein et al., 2021). A human pose is defined as highly ambiguous if at least one fitted Gaussian has a width greater than 5 pixels, which is caused by occluded joints or complex poses. On MPI-INF-3DHP, we follow DiffPose (Holmquist & Wandt, 2023) by directly applying the model trained on Human3.6M to the test set without any extra training or fine-tuning. Lastly, on 3DPW, we use the training and testing protocol from Marcard et al. (2018).

### 4.2 Metrics

For all datasets, we follow standard protocols of human pose estimation. The first protocol measures the Euclidean distance between the root-aligned predicted poses and the corresponding ground truths, often known as *Mean Per-Joint Position Error* (MPJPE). The second protocol also calculates the MPJPE, but with the *Procrustes alignment* beforehand, namely P-MPJPE. Procrustes alignment translates, rotates and scales the predicted 3D pose to the ground-truth pose, eliminating the global orientation and scaling factor from the measurement and sorely focus on the estimated shape and pose (Ionescu et al., 2014). We follow the established protocol (Wehrbein et al., 2021; Ci et al., 2023; Li et al., 2022a; Holmquist & Wandt, 2023) by computing the minimum MPJPE from a set of H hypotheses. We also report the *Percentage of Correct Keypoints* (PCK), measuring the percentage of joint predictions that are within a distance of 150mm compared to their ground-truth locations. PCK is also the main evaluation metric for the MPI-INF-3DHP (Mehta et al., 2017a). Following Wandt et al. (2021), we compute the *Correct Poses Score* (CPS) that only considers predicted poses to be correct if all joint errors fall below a threshold. Unlike PCK, to be independent of the threshold selection, the CPS measures an area under the curve for a range of thresholds from 0mm to 300mm. In all metrics, to effectively illustrate the reproducibility of FMPose, we report the average results collected from five random seeds from 40 to 44.

### 4.3 Comparison to related works

As the main contribution of FMPose is the introduction of the OT-path flow matching for multi-hypothesis 3D human pose generation, in this section, we report two different approaches with which we can obtain multiple hypotheses in FMPose using the heatmaps estimated by HRNet: a) selecting top-$k$ arguments and b) employing the random sampling strategy from DiffPose (Holmquist & Wandt, 2023). Essentially, this choice determines the input to the GCN module (see Figure 1), producing the condition vector $c$ in Equation (4). The chosen ODE steps for the main experiments is 25, see Section 4.4.4 for more information.

Table 1: Quantitative results on the Human3.6M dataset. Single: methods that only use single-frame input. H: the number of 3D pose hypotheses. **Bold:** the best evaluation results compared to other single-image probabilistic 3D HPE methods. †GFPose Ci et al. (2023) does not follow the standard data processing protocol and relies on the ground-truth information to obtain their 3D human pose estimations at inference, leading to incomparable results.

| Method | Single | H | MPJPE↓ | P-MPJPE↓ |
|---|---|---|---|---|
| VideoPose3D (Pavllo et al., 2019) | - | 1 | 46.8 | 36.5 |
| PoseFormer (Zheng et al., 2021) | - | 1 | 44.3 | 34.6 |
| MHFormer (Li et al., 2022a) | - | 3 | 43.0 | 34.4 |
| ManiPose (Rommel et al., 2024) | - | 5 | 39.1 | 34.1 |
| Martinez et al. (2017) | ✓ | 1 | 62.9 | 47.7 |
| Wehrbein et al. (2021) | ✓ | 1 | 61.8 | 43.8 |
| GFPose† Ci et al. (2023) | ✓ | 1 | 67.7 | 53.7 |
| DiffPose (Holmquist & Wandt, 2023) | ✓ | 1 | 64.5 | 45.2 |
| FMPose$_{random}$ (Ours) | ✓ | 1 | $62.3 \pm 0.5$ | $44.0 \pm 0.2$ |
| FMPose$_{top\text{-}k}$ (Ours) | ✓ | 1 | $58.9 \pm 0.4$ | $41.5 \pm 0.1$ |
| Li & Lee (2019) | ✓ | 5 | 52.7 | 42.6 |
| Li & Lee (2020) | ✓ | 10 | 73.9 | 44.3 |
| Sharma et al. (2019) | ✓ | 10 | 46.8 | 37.3 |
| GraphMDN Oikarinen et al. (2021) | ✓ | 200 | 46.2 | 36.3 |
| Wehrbein et al. (2021) | ✓ | 200 | 44.3 | 32.4 |
| GFPose† Ci et al. (2023) | ✓ | 200 | 35.8 | 30.6 |
| DiffPose Holmquist & Wandt (2023) | ✓ | 200 | $44.2 \pm 0.2$ | $32.1 \pm 0.1$ |
| FMPose$_{random}$ (Ours) | ✓ | 200 | $42.6 \pm 0.3$ | $31.7 \pm 0.1$ |
| FMPose$_{top\text{-}k}$ (Ours) | ✓ | 200 | $\mathbf{41.7} \pm 0.3$ | $\mathbf{30.6} \pm 0.1$ |

### 4.3.1 Human3.6M

First, we conduct the standard evaluation protocol of Human3.6M by testing the model predictions on every 64$^{\text{th}}$ frame from the action sequences. The results are demonstrated in Table 1 in comparison to the related works. On average, our FMPose$_{top\text{-}k}$ outperforms all other comparable methods on the two widely reported metrics: MPJPE and P-MPJPE. Compared to the direct competitor DiffPose, FMPose achieves a 2.5 mm (5.6%) improvement on the average MPJPE and 1.5 mm (4.6%) on the P-MPJPE. Additionally, using the same deterministic initial sample at zero ($H = 1$), FMPose$_{top\text{-}k}$ provides more accurate 3D pose estimation than, *e.g.*, Wehrbein et al. (2021) and DiffPose. Specifically, compared to DiffPose, we reduce 5.6 mm (8.7%) in MPJPE and 3.7 mm (8.1%) in P-MPJPE. While the results from GFPose Ci et al. (2023) with $H = 200$ appear to be better, they are incomparable since GFPose makes predictions in pixel space and requires ground-truth root translation, perspective ratio, and camera intrinsics to recover the 3D pose in camera space. Adapting GFPose's code to follow the standard protocol causes a complete collapse in its performance; thus, we gray out GFPose to indicate that it is not comparable.

Secondly, we report the evaluation results of FMPose on the highly ambiguous poses from the Human3.6M dataset defined by Wehrbein et al. (2021), in Table 2. With the same random sampling strategy extracting from 2D heatmap, FMPose$_{random}$ outperforms the current SOTA, DiffPose, by 7.6 mm (11.4%), 1.8 mm (3.7%) in P-MPJPE, 2.4% in PCK, while having the same performance on CPS. The significant improvement in the MPJPE is a direct result of using the proposed continuous probability flows, which effectively map the Gaussian distribution towards the plausible 3D pose distribution via the OT path, unlike the more challenging stochastic path of the diffusion models. The similar results in CPS signify the similar uncertainty coverage to highly ambiguous poses of FMPose$_{random}$ compared to DiffPose, while having more accurate 3D joint predictions (MPJPE). Despite lower CPS, the FMPose$_{top\text{-}k}$ still achieves higher performance than DiffPose by 6.5 mm (9.8%) in MPJPE and 1 mm (2%) in P-MPJPE.

Table 2: Quantitative results on the hard samples of Human3.6M, defined by Wehrbein et al. (2021). Our FMPose$_{random}$ significantly outperforms all related methods, especially on the MPJPE metric.

| Method | MPJPE ↓ | P-MPJPE ↓ | PCK ↑ | CPS ↑ |
|---|---|---|---|---|
| Li & Lee (2019) | 81.1 | 66.0 | 85.7 | 119.9 |
| Sharma et al. (2019) | 78.3 | 61.1 | 88.5 | 136.4 |
| Wehrbein et al. (2021) | 71.0 | 54.2 | 93.4 | 171.0 |
| DiffPose (Holmquist & Wandt, 2023) | $66.5 \pm 1.4$ | $48.5 \pm 0.2$ | $94.3 \pm 0.1$ | $194.2 \pm 2.2$ |
| FMPose$_{random}$ (Ours) | $\mathbf{58.9 \pm 0.1}$ | $\mathbf{46.7 \pm 0.2}$ | $\mathbf{96.6 \pm 0.1}$ | $\mathbf{194.7 \pm 0.9}$ |
| FMPose$_{top\text{-}k}$ (Ours) | $60.0 \pm 0.3$ | $47.5 \pm 0.3$ | $96.0 \pm 0.1$ | $187.6 \pm 1.6$ |

Table 3: Quantitative PCK results on the MPI-INF-3DHP dataset. [†]GFPose contains non-comparable results (see the explanation in the text). With respect to other comparable methods, FMPose achieves better results across different setups, demonstrating the high generalizability to unseen data.

| Method | In. GS↑ | In. no GS↑ | Out.↑ | All↑ |
|---|---|---|---|---|
| Li & Lee (2019) | 70.1 | 68.2 | 66.6 | 67.9 |
| Li & Lee (2020) | 86.9 | **86.6** | 79.3 | 85.0 |
| Wehrbein et al. (2021) | 86.6 | 82.8 | 82.5 | 84.3 |
| GFPose[†] (Ci et al., 2023) | 88.4 | 87.1 | 84.3 | 86.9 |
| DiffPose (Holmquist & Wandt, 2023) | $87.4 \pm 0.4$ | $82.5 \pm 0.1$ | $83.3 \pm 0.2$ | $84.6 \pm 0.1$ |
| FMPose (Ours) | $\mathbf{87.9 \pm 0.3}$ | $83.3 \pm 0.2$ | $\mathbf{84.1 \pm 0.2}$ | $\mathbf{85.3 \pm 0.3}$ |

Using the random sampling strategy from DiffPose (Holmquist & Wandt, 2023) helps widen the coverage of pose uncertainty, as demonstrated by the superior performance of FMPose$_{random}$ in Table 2. However, taking too many lower-confidence heatmap arguments into account can worsen the performance of the model on more certain poses, resulting in inferior results of FMPose$_{random}$ compared to FMPose$_{top\text{-}k}$ on the standard evaluation setting in Table 1. Therefore, there exists a trade-off between the two strategies that can be easily alternated for FMPose depending on the needs of any downstream task: top-$k$ is a more deterministic approach that ensures highly accurate 3D estimations, while random sampling helps capturing low-confidence human joints in highly ambiguous poses. Considering the trade-off, we select the top-$k$ version of FMPose as our main model for further evaluations on the remaining datasets.

### 4.3.2 MPI-INF-3DHP

To demonstrate the generalization of the proposed FMPose, we conduct evaluation on the test set of MPI-INF-3DHP without additional training or fine-tuning, similar to DiffPose. The quantitative results are reported in Table 3, demonstrated via four different setups: indoor with green screen (In. GS), indoor without green screen (In. no GS), outdoor (Out.) and the average of all scenarios. FMPose achieves competitive PCK evaluation score with respect to comparable methods. Specifically, our FMPose outperforms the closest competitor DiffPose on all PCK setups. We record state-of-the-art performance on the indoor with green screen (PCK of 87.9%) and the more challenging outdoor samples (PCK of 84.1%).

### 4.3.3 3DPW

We conduct experiments on the more challenging 3DPW dataset and report the results in Table 4. Note that the baselines (Li et al., 2021; Goel et al., 2023; Li et al., 2022b; Baradel et al., 2024) are mesh-based methods, which could not be directly compared with the joint-based method FMPose, and the result in Table 4 is only for reference. Inspired by the setup of recent work (Baradel et al., 2024; Chen et al., 2025), we report both the FMPose trained from scratch on the 3DPW training set, and the fine-tuned version initialized from the pre-trained model on Human3.6M. FMPose achieves the lowest joint-wise distance error (MPJPE)

Table 4: Quantitative results on the 3DPW dataset. FMPose significantly outperforms other single-frame methods, while staying competitive with multi-frame approaches.

| Method | Single | MPJPE | P-MPJPE |
|---|---|---|---|
| MotionBERT (Zhu et al., 2023) | - | 68.8 | 40.6 |
| WHAM (Shin et al., 2024) | - | 57.8 | 35.9 |
| ExtPose (Chen et al., 2025) | - | 54.2 | 34.0 |
| HybrIK (Li et al., 2021) | ✓ | 74.1 | 45.0 |
| HMR2.0 (Goel et al., 2023) | ✓ | 70.0 | 44.5 |
| CLIFF (Li et al., 2022b) | ✓ | 69.0 | 43.0 |
| MultiHMR (Baradel et al., 2024) | ✓ | 61.4 | 41.7 |
| FMPose$_{\text{scratch}}$ (Ours) | ✓ | $63.9 \pm 0.3$ | $42.3 \pm 0.2$ |
| FMPose$_{\text{finetuned}}$ (Ours) | ✓ | $\mathbf{56.2} \pm 0.2$ | $\mathbf{35.4} \pm 0.1$ |

Table 5: Ablation studies on the design of FMPose. Param. is the total number of training parameters. P.time is the processing time of FMPose in a single pass with 200 hypotheses. To verify the contribution of each component, we individually remove them from the model and train it from scratch with different seeds. The full design of FMPose demonstrates high performance gains compared to other configurations, with minimal computational overhead.

| FMPose | Param. | P.time | Human3.6M | | Ambiguous Human3.6M | |
|---|---|---|---|---|---|---|
| | | | MPJPE↓ | P-MPJPE↓ | MPJPE↓ | P-MPJPE↓ |
| w/o condition | 4.3M | 28ms | $122.2 \pm 0.2$ | $72.3 \pm 0.1$ | $169.2 \pm 0.1$ | $96.1 \pm 0.1$ |
| w/o GCN | 4.6M | 28ms | $44.0 \pm 0.3$ | $31.6 \pm 0.1$ | $63.0 \pm 0.3$ | $49.5 \pm 0.5$ |
| w/o dropout | 4.5M | 37ms | $43.9 \pm 0.2$ | $31.4 \pm 0.1$ | $63.0 \pm 0.4$ | $49.4 \pm 0.2$ |
| w/o top-$k$ | 4.5M | 34ms | $43.8 \pm 0.3$ | $32.1 \pm 0.2$ | $62.2 \pm 0.4$ | $48.9 \pm 0.5$ |
| w/o learn. $A$ | 4.5M | 36ms | $42.5 \pm 0.2$ | $30.8 \pm 0.1$ | $61.1 \pm 0.5$ | $47.7 \pm 0.5$ |
| w. transformer | 4.5M | 75ms | $43.0 \pm 0.3$ | $30.9 \pm 0.2$ | $61.3 \pm 0.5$ | $47.7 \pm 0.3$ |
| Full (ours) | 4.5M | 36ms | $\mathbf{41.7} \pm 0.3$ | $\mathbf{30.5} \pm 0.1$ | $\mathbf{59.8} \pm 0.3$ | $\mathbf{47.2} \pm 0.2$ |

compared to other single-image 3D pose estimation methods. We improve over the closest competitor, MultiHMR (Baradel et al., 2024), by 8.4% on MPJPE and 15.1% on P-MPJPE. Compared to multi-image methods such as WHAM (Shin et al., 2024) or ExtPose (Chen et al., 2025), FMPose maintains competitive performance while being a single-frame method.

## 4.4 Ablation studies

We conduct several ablation studies to verify our contributions and to evaluate FMPose in multiple different settings. For ablation, we follow the same evaluation protocol for Human3.6M presented in Section 4.1.

### 4.4.1 Lifting condition

The condition tensor $c$ in Equation (4) is a vital element to realize the task of 2D-3D human pose estimation. Without the condition, it is impossible to reconstruct the 3D poses from the input Gaussian noises. A demonstration is given in Table 5: *FMPose w/o condition* results in very low evaluations across all metrics. Besides, traditionally, the lifting condition is simply the maximum argument of the predicted heatmap, taking into account only the most probable joint location. We show in Table 5 that this approach (*FMPose w/o top-k*) fails to also consider the possibility of less confident joint locations, leading to a decrease of 2.1 mm (4.8%) in MPJPE and 2.4 mm (3.9%) in ambiguous MPJPE, compared to our full model. In the architecture of *FMPose w. transformer*, we replace the GCN module with the transformer-based conditioning similar to DiffPose, while maintaining the same number of learning parameters of 4.5M. Our full FMPose

Table 6: Ablation studies for different ODE solvers. P.time is the total processing time for the FMPose to generate 200 hypotheses. Higher-order solvers consume more processing time, and the RK2 solver achieves the best trade-off between accuracy and computational cost, thus selected as the main solver for FMPose.

| FMPose | P.time | MPJPE↓ | P-MPJPE↓ | PCK↑ | CPS↑ |
|---|---|---|---|---|---|
| w. RK1 | 20ms | $43.1 \pm 0.2$ | $31.0 \pm 0.2$ | $98.9 \pm 0.0$ | $234.2 \pm 0.6$ |
| w. RK2 | 36ms | $41.7 \pm 0.2$ | $\mathbf{30.4} \pm 0.2$ | $\mathbf{99.1} \pm 0.0$ | $237.2 \pm 0.4$ |
| w. RK3 | 55ms | $\mathbf{41.6} \pm 0.2$ | $30.5 \pm 0.1$ | $99.1 \pm 0.1$ | $\mathbf{237.3} \pm 0.3$ |
| w. RK4 | 71ms | $42.3 \pm 0.1$ | $30.9 \pm 0.1$ | $99.1 \pm 0.1$ | $237.0 \pm 0.4$ |

Table 7: Ablation study on the choice of ODE solving steps. Speed: the speed to generate 200 hypotheses per input image. **Bold**: the best results and Underlined: the second best results. Higher number of steps increase the accuracy, while requiring more computational time. The best trade-off is at 25 solving steps.

| RK2 | | Human3.6M | | Ambiguous Human3.6M | |
|---|---|---|---|---|---|
| Steps | P.time | MPJPE↓ | P-MPJPE↓ | MPJPE↓ | P-MPJPE↓ |
| 5 | 7.6ms | $44.2 \pm 0.2$ | $32.0 \pm 0.1$ | $63.1 \pm 0.7$ | $49.6 \pm 0.4$ |
| 10 | 14.7ms | $42.3 \pm 0.3$ | $30.8 \pm 0.1$ | $60.5 \pm 0.5$ | $47.8 \pm 0.4$ |
| 15 | 22.2ms | $41.9 \pm 0.3$ | $30.6 \pm 0.1$ | $59.9 \pm 0.4$ | $47.2 \pm 0.3$ |
| 20 | 28.7ms | $41.8 \pm 0.3$ | $\underline{30.5} \pm 0.1$ | $\underline{59.8} \pm 0.4$ | $\mathbf{47.1} \pm 0.2$ |
| 25 | 36.1ms | $\mathbf{41.7} \pm 0.2$ | $\mathbf{30.5} \pm 0.1$ | $\mathbf{59.8} \pm 0.3$ | $\underline{47.2} \pm 0.3$ |
| 30 | 42.7ms | $\underline{41.7} \pm 0.3$ | $30.5 \pm 0.1$ | $59.9 \pm 0.4$ | $47.4 \pm 0.3$ |
| 35 | 48.3ms | $41.7 \pm 0.2$ | $30.6 \pm 0.1$ | $60.0 \pm 0.4$ | $47.4 \pm 0.3$ |
| 40 | 56.7ms | $41.8 \pm 0.2$ | $30.7 \pm 0.1$ | $60.1 \pm 0.3$ | $47.7 \pm 0.3$ |

has lower MPJPE of $1.3\,\mathrm{mm}$ (3.0%) and ambiguous MPJPE of $1.5\,\mathrm{mm}$ (2.5%), demonstrating the advantage of the GCN conditioning compared to the transformer-based conditioning.

### 4.4.2   Graph features extraction

We verify the contribution of the GCN by replacing it with a fully connected layer, similar to Wehrbein et al. (2021). A simple learning architecture as the fully connected layer, referred to as *FMPose w/o GCN* in Table 5, cannot capture the complexity of the 2D pose for the lifting task, resulting in a $2.3\,\mathrm{mm}$ (5.2%) increase in MPJPE, compared to the full model. Furthermore, the adjacency matrix $A$ of the GCN, which encodes the connections between human joints, can be learned during training. While the entries of $A$ can be assigned based on the natural connection of the human body, demonstrated as *FMPose w/o learn A*, we found that learning from scratch (the full model) leads to a decrease of $0.8\,\mathrm{mm}$ (1.9%) in MPJPE. Therefore, the proposed GCN effectively extracts features from the 2D heatmaps for the lifting condition.

### 4.4.3   Inference with CNF

Unlike the discrete mapping as in Wehrbein et al. (2021) and DiffPose (Holmquist & Wandt, 2023), FMPose is a continuous function that can be approximated using a numerical solver. The choice of solver and its configuration is also an important aspect of the proposed approach. The quantitative results by using different numerical solvers can be seen in Table 6. Compared to other solvers, the RK2 achieves the best trade-off between prediction accuracy and processing speed, with 41.7mm in MPJPE and 36ms (28FPS) to generate 200 hypotheses. Therefore, the RK2 is selected to be the main ODE solver of FMPose.

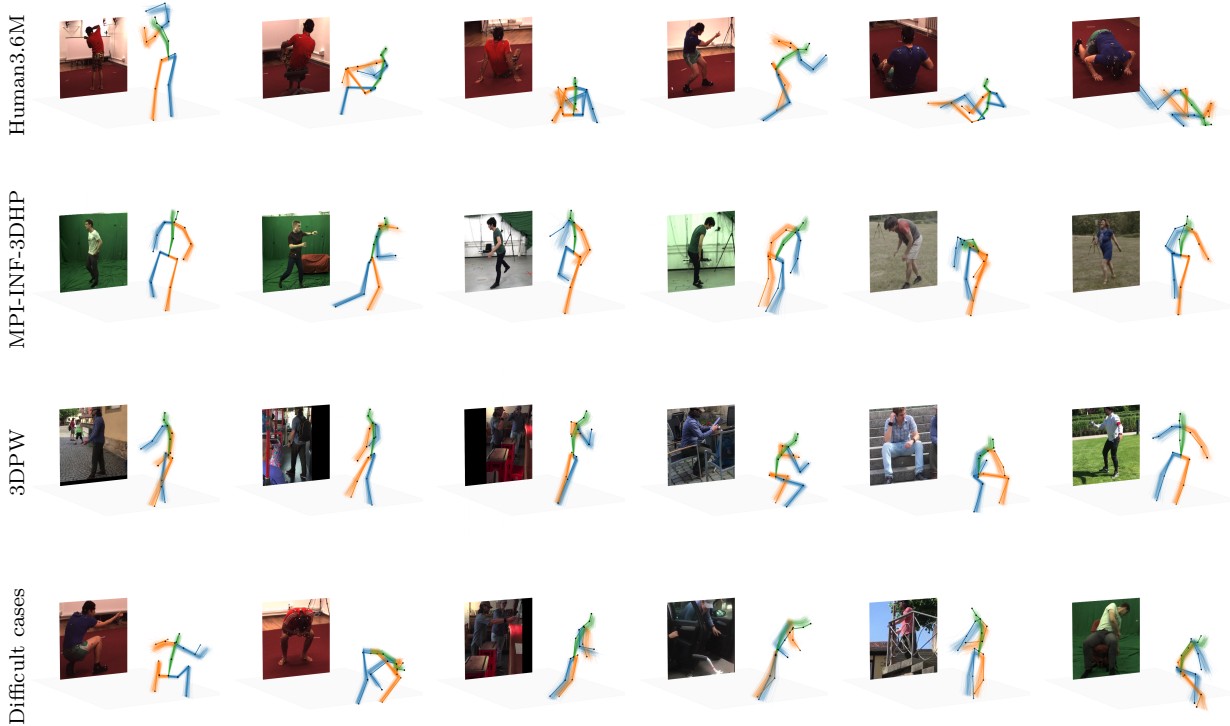

Figure 2: Qualitative results of FMPose on the Human3.6M, MPI-INF-3DHP and 3DPW datasets. The human pose contains 17 joints, with the right side encoded with blue color, left side with orange, and the middle with green. All 200 hypotheses are drawn for each example, with the best hypothesis in bold. The examples are randomly selected to cover all scenarios: indoor, complex background and outdoor. It is visually clear that FMPose performs well on different setups. On challenging scenarios, i.e. fully occluded joints from the camera view, FMPose produces a wider range of hypotheses along the optical axis, providing the additional uncertainty awareness for downstream tasks.

#### 4.4.4 Number of steps in ODE solver

Besides the choice of ODE solver in Table 6, the number of discrete integration time steps also plays an important role in ensuring the best 3D estimations. We present an ablation study on a different number of total time steps for the selected RK2 solver in Table 7. Note that, since we assume the probability mapping from the Gaussian distribution to the 3D pose distribution is bounded in $[0, 1]$, the corresponding step size for each choice of ODE steps is $\Delta t = 1/\text{steps}$. Generally, a higher number of steps with a smaller step size ensures the stability and accuracy of the ODE solver, but introduces computational overhead. From Table 7, the ODE performance saturates at 25 steps; further increments result in no gains while significantly increasing the computational time. Therefore, we select the total number of steps to be 25 for FMPose.

### 4.5 Complexity comparison to DiffPose

As demonstrated in Tables 1 to 3, FMPose consistently outperforms the SOTA diffusion-based method DiffPose (Holmquist & Wandt, 2023) across different evaluation settings. In this section, we provide a comparative complexity study between our FMPose and DiffPose in Table 8. Based on the official code, we record that the total number of parameters of DiffPose is 11.5M, while our FMPose has a significantly lower number of parameters with only 4.5M (60% lower). Besides the similar neural network $f_\theta$ for flow velocity estimation in FMPose and for denoising process in DiffPose, the significant higher complexity in DiffPose comes from the transformer module for extracting its 2D-3D lifting condition, with 5M parameters and consumes 1.9ms per pass, while our GCN has only 162K parameters and 0.16ms per pass (91.5% faster).

Table 8: Computational complexity comparisons between FMPose and DiffPose using an NVIDIA A40. Param. is the total number of parameters. Steps is the number of ODE solving steps in FMPose and the number of denoising steps in DiffPose, respectively. The final steps are from the models reported in the main paper. The MPJPE results are the evaluation on the ambiguous Human3.6M.

| Method | Param | Steps | P.time | MPJPE | P-MPJPE |
|--------|-------|-------|--------|-------|---------|
| DiffPose | 11.5M | 32 (final) | 17.5ms | $66.5 \pm 1.4$ | $48.5 \pm 0.2$ |
| FMPose | 4.5M | 5 | **7.6**ms | $63.1 \pm 0.7$ | $49.6 \pm 0.4$ |
| FMPose | 4.5M | 7 | 10.5ms | $61.8 \pm 0.4$ | $48.5 \pm 0.3$ |
| FMPose | 4.5M | 10 | 14.7ms | $60.5 \pm 0.5$ | $47.8 \pm 0.4$ |
| FMPose | 4.5M | 15 | 22.2ms | $59.9 \pm 0.4$ | **47.2** $\pm 0.3$ |
| FMPose | 4.5M | 25 (final) | 36.1ms | **59.8** $\pm 0.3$ | **47.2** $\pm 0.3$ |

From Table 8, we choose the FMPose with 25 solving steps as the best performing model, with a speed of 28 FPS. However, depending on the computational needs of the downstream tasks, FMPose's configuration can easily be adjusted to a lower number of solving steps. Compared to DiffPose with 32 denoising steps in 17.5ms, our FMPose with only 7 steps can achieve a lower processing time of 10.5 mm (40% faster), while having equal P-MPJPE and considerable lower MPJPE. The FMPose with only 10 steps is 16.0% faster and has 9.0% lower MPJPE than DiffPose, highlighting the advantage of the stable optimal transport flow matching in generating higher-quality hypotheses with fewer ODE steps. Please visit our Appendix C for the additional comparison on pose hypothesis concentration level between FMPose and DiffPose.

## 4.6 Qualitative results

We visualize some outputs of FMPose on the three datasets in Figure 2. For each image input, FMPose produces a set of 200 hypotheses of 3D human pose, with the best one being amplified in color. Clearly visible poses contain much lower divergence in hypothesis generations due to highly confident estimations. For body joints that is occluded from the camera view, i.e. the third example of MPI-INF-3DHP, FMPose creates a much wider hypothesis generation, effectively model the uncertainty of the right hand joint along the optical axis. Furthermore, when the scene is fully obstructed with objects and the 2D detector partially fails, i.e. third and forth examples of the difficult cases, our method can still produce plausible 3D poses (upright standing) because it is trained with a robust optimal probability flow towards the true 3D pose distribution. For more qualitative results on in-the-wild examples taken from Johnson & Everingham (2011); Kazemi et al. (2013) with no further training required, please visit our Appendix D.

## 5 Conclusion

We introduced a novel approach for probabilistic monocular 3D human pose estimation, FMPose, a continuous normalizing flow generative model trained with the optimal transport path. FMPose outperforms other related methods by a large margin in 3D joint estimation accuracy, while being easy to train via the flow matching framework. We also propose using the GCN for extracting the 2D-3D lifting condition from the 2D heatmaps that naturally consider the learnable relations between human joints. Being a probabilistic method, FMPose can benefit downstream tasks by providing accurate uncertainty measurements about the occluded joints or challenging human poses. Under equal-accuracy configurations, FMPose with optimal transport provides a significantly faster processing speed compared to diffusion-based methods.

**Limitations and future work** Besides the 2D keypoint information, the interaction between the human body with the surrounding environment could also play an important role in disambiguating the challenging scenarios. FMPose currently bypasses all this information and uses only the 2D heatmaps provided by the joint detector, and failed 2D detections would decrease the performance of our model. Developing a new method that directly utilizes the input image with human-scene interactions, most importantly, contact estimation, is a worthwhile continuation of the research direction for the paper.

**Acknowledgments and Disclosure of Funding**   This research is supported by the Wallenberg Artificial Intelligence, Autonomous Systems and Software Program (WASP), funded by Knut and Alice Wallenberg Foundation. The computational resources were provided by the National Academic Infrastructure for Supercomputing in Sweden (NAISS) at C3SE, and by the Berzelius resource, provided by the Knut and Alice Wallenberg Foundation at the National Supercomputer Centre.

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
