## Appendix of FMPose

### A    Additional ablation on top-$k$ selection

The top-$k$ determines how many top arguments from the 2D heatmaps are used as inputs for the GCN of FMPose. We demonstrate the ablation on different values of $k$ in Table 9. Due to the limited computational resources, we report the error bars across only three random seeds: 40, 42, and 44. Furthermore, to prevent the network from overfitting to a few highest-confidence arguments, we randomly shuffle the top-$k$ arguments before passing them as inputs. Random shuffle is applied to all the experiments in the main paper and the supplementary. The results in Table 9 show that using a higher $k$ improves the 3D joint estimation accuracy on the indoor dataset Human3.6M, but reduces the generalization to the more challenging outdoor dataset MPI-INF-3DHP. We found the optimal $k$ to be at 48, where the trade-off in performance is the best between the two datasets. All the experiments in the main paper are thus conducted with $k = 48$.

Table 9: Ablation study on the choice of top-$k$ arguments for FMPose. (A)MPJPE: MPJPE on the ambiguous Human3.6M set. (O)PCK: PCK on the outdoor subset of MPI-INF-3DHP. **Bold**: the best results and Underlined: the second best results. The top-$k$ at 48 achieves the best performance trade-off.

| | Human3.6M | | MPI-INF-3DHP | |
| $k$ | MPJPE↓ | (A)MPJPE↓ | PCK↑ | (O)PCK↑ |
|---|---|---|---|---|
| 8 | 42.5±0.0 | 61.3±0.6 | 85.5±0.1 | 84.4±0.3 |
| 16 | 42.0±0.3 | 60.5±0.3 | **85.6**±0.2 | **84.6**±0.3 |
| 32 | 42.0±0.3 | 60.2±0.6 | 85.5±0.1 | 84.4±0.2 |
| **48** | 41.8±0.1 | 60.2±0.3 | 85.4±0.2 | 84.4±0.2 |
| 64 | 41.8±0.0 | **59.9**±0.5 | 85.1±0.3 | 84.2±0.4 |
| 80 | **41.5**±0.4 | 60.4±0.5 | 84.8±0.2 | 83.2±0.4 |
| 96 | 42.0±0.1 | 60.5±0.4 | 84.1±0.2 | 81.4±0.3 |

### B    Adjacency matrix

As described in Sec. 3.2 in the main paper, the adjacency matrix $A$ is adaptively learned together with FM-Pose, initialized from zeros. We visualize the learned adjacency matrix $A$ from the Human3.6M dataset (left) and the 3DPW dataset (right) in Figure 3. Top five correlations from the matrix learned from Human3.6M are recorded at: Left hip – Neck, Right knee – Nose, Thorax – Elbow, Right hip – Head, Right elbow – Left hip, and these correlations are beneficial towards the 3D pose generations. Interestingly, there is not much shifts in top five correlations recorded, the only change between the learned adjacency from 3DPW to Human3.6M is the 5[th] entry, shifting from Right elbow – Left to Left wrist – Left hip. This highlights the strong generalizability of FMPose, as its performance is not contingent on overfitting to any specific dataset.

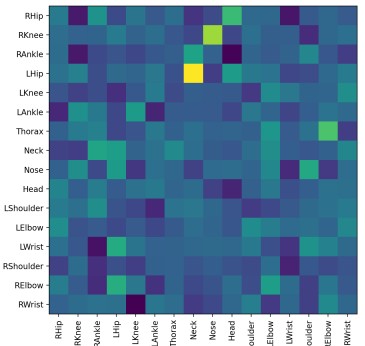 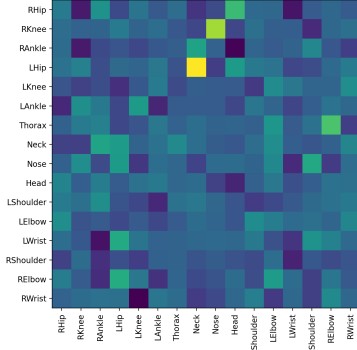

Figure 3: Visualization of the adjacency matrix learned from the Human3.6M (left) and the 3DPW (right) datasets. The brighter the entry, the more correlation between the two joints towards the 3D pose generations.

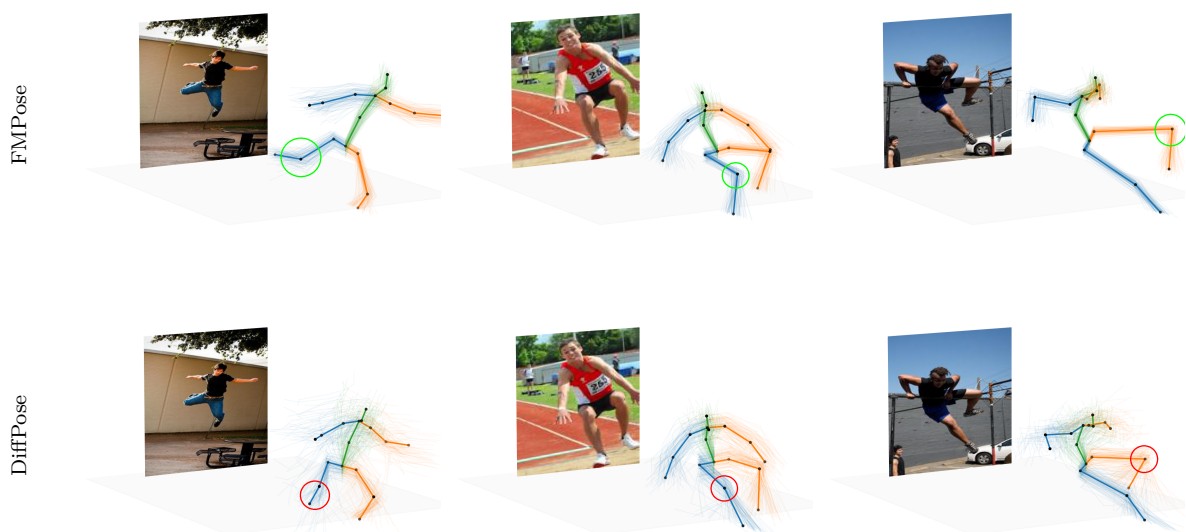

Figure 4: Qualitative comparisons between FMPose and DiffPose on three challenging samples taken from the Leeds Sport Pose dataset. The total number of drawn hypotheses for both methods is 200 with the mean pose highlighted in higher intensity. FMPose produces more plausible poses (green circles) than DiffPose (red circles) on challenging scenarios.

Table 10: Average standard deviations (measured in mm) of 200 hypotheses of 3D human pose on three out-of-domain datasets: MPI-INF-3DHP, KTH Football, and Leeds Sport Pose. Lower standard deviation means higher concentrated hypotheses. Results are averaged over 5 random seeds.

| Method | MPI-INF-3DHP | KTH Football | Leeds Sport Pose | Average |
|---|---|---|---|---|
| DiffPose (Holmquist & Wandt, 2023) | $95.1 \pm 12.4$ | $58.2 \pm 3.5$ | $58.2 \pm 2.6$ | 70.5 |
| FMPose (Ours) | $35.3 \pm 1.6$ ↓ 62.8% | $25.9 \pm 1.9$ ↓ 55.5% | $26.2 \pm 1.2$ ↓ 55.0% | 29.1 ↓ 58.7% |

## C  Additional comparisons to DiffPose

To further highlight the different between FMPose and DiffPose, we present additional qualitative results of both methods on the Leed Sport Pose dataset in Figure 4. As shown in Figure 4, FMPose produces more plausible poses with respect to the input image compared to DiffPose. The example in the first column shows a leaping motion, where FMPose predicts correctly with the right leg raising backward, whereas DiffPose predict a forward knee-bending pose. Additionally, we present in Table 10 the average standard deviations of the estimated body joints on three datasets: MPI-INF-3DHP (Mehta et al., 2017a), KTH Football (Kazemi et al., 2013) and Leeds Sport Pose (Johnson & Everingham, 2011). These three out-of-domain datasets are not exposed to the NN model during training. Hypotheses generated by FMPose has significantly lower standard deviation than DiffPose, up to 58.7% on average. Based on the results from Figure 4 and Table 10, we observe that the hypotheses estimated by FMPose are more concentrated around the mean pose than the 200 hypotheses of DiffPose, which highlights the advantage of the deterministic OT flows over the stochastic diffusion model in the task of 2D-3D pose lifting.

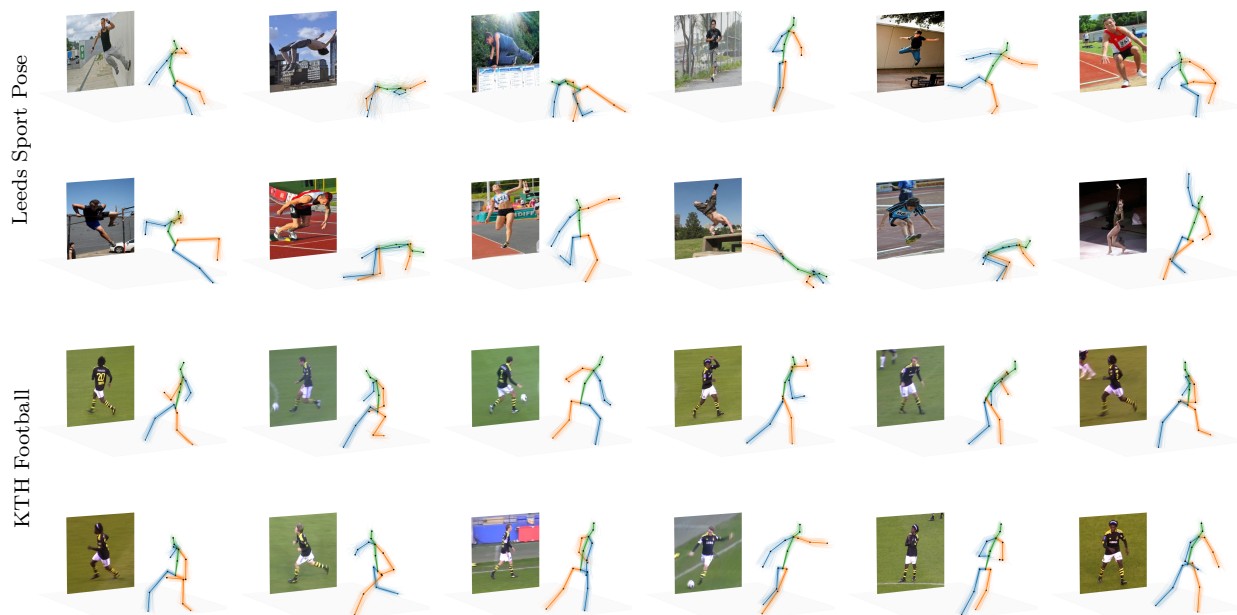

Figure 5: Additional qualitative results on in-the-wild human poses. First two rows are examples taken from the Leeds Sport Pose dataset (Johnson & Everingham, 2011), consisting of complex sport poses in a wide variety of background environments. The next two rows are examples from the KTH Football dataset (Kazemi et al., 2013), which mostly contains footage of football players playing on the field, with green grass. All 200 hypotheses are visualized, with the mean pose is bold. In both cases, FMPose performs well with plausible 3D reconstructions.

## D  More qualitative results

We provide additional qualitative results in Figure 5, on two in-the-wild datasets: the Leeds Sport Pose (Johnson & Everingham, 2011) and the KTH Football (Kazemi et al., 2013). These datasets is collected in the wild, no ground-truth 3D pose is available for quantitative evaluation, thus only qualitative results are presented. The pre-trained model on the Human3.6M dataset is applied to the images without any extra fine-tuning. FMPose performs well on difficult sport poses from various scenes. In abnormal challenging case such as the flipping motion in second image of the first row, the 200 hypotheses drawn by FMPose significantly diverge to cover the high uncertainty, yet the mean pose (bold) is still plausible with respect to the input.