# OpenReview forum: "Flow Matching for Probabilistic Monocular 3D Human Pose Estimation"
_TMLR — Accepted by TMLR_

### Review · Reviewer_Af5w · 2026-02-22

**Summary Of Contributions:**

The paper proposes FMPose, a conditional generative flow-matching approach to 3D human pose estimation, where the conditioning is based on 2D inputs. The authors process 2D heatmaps into $k$ poses with pre-trained HRNet, over which GCNs are employed to extract features that are used as conditioning to the mapping between noise and poses. The model is evaluated on three human pose estimation benchmarks where it performs favorably to the baselines, and also seems to be faster than previous SOTA.

**Additional Comments:**

**Strengths**:
* Motivation: the positioning of this work is well-motivated as good 3D pose estimation may be beneficial to other domains.
* Rigorous experiments and analysis.
* Strong performance over baselines.
* The method seems to be fast and operate at high FPS.

**Weaknesses**:
* The novelty is rather incremental in the sense that the authors change the training objective to flow-matching  from diffusion.
* Limitations discussion could be extended.

**Audience:**

Yes

**Audience Explanation:**

Yes, the community of 3D pose estimation and domains that build upon accurate 3D pose estimation would be interested in the findings.

**Claims And Evidence:**

Yes

**Claims Explanation:**

Yes, the claims are supported by clear evidence. See below for strengths.

**Requested Changes:**

* I think “Implementation Details” should come before “Experiments”.
* I did not understand how $c$ is integrated in $f$. Is it simple concatenation?
* What is the number of ODE steps used at inference time for the experiments? Is it 25? The number of steps is only mentioned in the ablation studies. This number should appear under the “Experiments” section or “Implementation Details”.

**Minor**:

* Figure 1 should include a block for HRNet, as it is not trivial to extract poses from images.
* Page 4, near Eq. 2: “...to a simple regression task based on Equations..” - missing "Equations" numbers.

---

> ### Author Response · Authors · 2026-04-30
> **Response to Reviewer Af5w**
>
> We thank the reviewer for their constructive feedback and the time spent evaluating our manuscript.
> We would like to provide our answers to the reviewer's requests in the following, and the manuscript has been updated accordingly.
>
> 1. Implementation details before Experiments
>
> Thanks for the suggestion! We have now moved the Implementation details to Section 3.4 in the Method section.
>
> 2. The integration of 2D condition $c$
>
> Yes, the 2D condition $c$ is simply concatenated with $x_t$ and $t$ to form the input tensor.
>
> 3. The final number of ODE steps at inference
>
> As discussed in the Section 4.4.4., the final number ODE steps for all the main experiments is $25$. We have now included clarifications about this information in the first paragraph of Section 4.3.
>
> 4. HRNet block in Figure 1
>
> We have now included an additional block for HRNet, taking RGB image as input and output the heatmaps.
>
> 5. Missing references
>
> We have now fixed the issue, thanks!

---

### Review · Reviewer_AAqs · 2026-03-04

**Summary Of Contributions:**

This paper proposes a generative framework for monocular 3D pose estimation, leveraging optimal transport flow matching to lift 2D-to-3D conditioning from a single image.

The main contributions are:

* Introducing optimal transport flow matching for 3D pose estimation.
* Incorporating graph neural networks to condition the flow-matching pipeline.
* Modeling multi-hypothesis predictions via uncertainty-aware conditioning, by passing the top-(k) conditioning candidates to the graph-based module.
* Achieving state-of-the-art performance, outperforming baselines across four evaluation metrics.

**Strengths:**
The paper is very well written and includes extensive experiments that support the claims, making it easy to follow. The experimental details are clearly described, which facilitates fair comparison and reproducibility for future work.

**Weaknesses:**
The paper lacks a diversity analysis of the 200 hypotheses generated by the generative models. Including such a comparison would provide additional insight into the variability and coverage of the generated poses.

**Additional Comments:**

Overall, this is a very well-written paper. However, I found the introduction somewhat lacking in coherence, and I would recommend revising it for better clarity and flow. That said, this is a minor issue, as most of the questions raised in the introduction are addressed later in the paper.

**Audience:**

Yes

**Audience Explanation:**

I think that, based on the literature in this domain, the use of optimal transport flow matching is particularly interesting. It introduces a new state-of-the-art approach that researchers in this field can use as a baseline and potentially build upon in future work. Moreover, the proposed model is relatively lightweight compared to DiffPose, which further increases its practical appeal.

**Claims And Evidence:**

Yes

**Claims Explanation:**

The paper presents a clear experimental setup across multiple datasets and baselines, demonstrating its superiority over prior methods on several metrics. It also provides strong qualitative visualizations showing how the model produces multiple hypotheses.

They further evaluate two uncertainty-aware conditioning strategies—random sampling and top-(k) selection—under two different experimental setups, including a more challenging one.

What I liked most is the detailed ablation study covering the key components of the pipeline, which clearly illustrates the contribution of each module to the overall performance.

**Requested Changes:**

**Critical comments**

1. I could not find any diversity metric for the 200 hypotheses generated by DiffPose and FMPose. In all reported metrics, only the best hypothesis seems to be selected. While the paper includes a figure illustrating hypotheses from FMPose, adding qualitative results for DiffPose would help highlight differences between the methods. In addition, reporting diversity statistics (such as the standard deviation) would provide intuition about how broad or concentrated the hypothesis set is for each model.

2. In the last line of Section 2.2, you state: “FMPose allows for straighter and better probability paths for generating high-quality 3D human pose hypotheses with fewer ODE steps.” Could you clarify what you mean by “better probability paths”? It is not obvious how “better” is defined for a probability path, and the statement currently feels subjective without a precise criterion.

3. In Section 4.4, you mention two conditioning approaches: top-(k) selection and random sampling (following DiffPose). However, it is unclear how DiffPose’s conditioning strategy works and how exactly you adapt it in your pipeline. Please elaborate on  the mechanism used in DiffPose, and why, based on your findings, these strategies better cover lower-confidence heatmaps.

4. Why is only one metric reported for some scenarios (e.g., MPI-INF-3DHP)? Please clarify the reason.

5. According to Table 8, it is surprising that FMPose with 25 steps is slower than DiffPose with 32 steps, especially since DiffPose is larger in terms of parameters and uses more steps. Could you explain this result in more detail ?

**Minor comments**

1. Consider explicitly stating that training is end-to-end, since the pipeline includes two modules (GCN and CNF) and this is an important implementation detail.

2. In Section 4.2 (“2D pose estimation”), you write: “resulting in an input tensor of shape (R^{J \times k}) for the GCN module.” I believe this should be (2k) (or otherwise clarify the dimensionality), since each joint typically has 2D coordinates per hypothesis.

3. The definition of P-MPJPE in the Metrics section is vague. Please provide a clearer definition and explain how it differs from MPJPE.

4. For Tables 3 and 4, it is unclear whether the results correspond to top-(k) conditioning or random sampling, since this is specified for other tables but not here. I suggest clarifying the conditioning strategy used for these tables.

---

> ### Author Response · Authors · 2026-04-30
> **Response to Reviewer AAqs**
>
> We thank the reviewer for their insightful comments and feedback. We updated the manuscript and would like to provide our answers to the reviewer's comments in the following.
>
> **Major comments**
>
> 1. Diversity analysis and additional qualitative results
>
> We add a Section D to the appendix to show the qualitative comparisons to DiffPose in Fig. 5.
> Additionally, in Tab. 10, we compute the average standard deviation over $200$ hypotheses for both FMPose and DiffPose on three datasets: MPI-INF-3DHP, KTH Football, and Leeds Sport Pose.
> The results show FMPose's hypothesis set is more concentrated than that of DiffPose, having a significantly lower average standard deviation by approximately $58.7$\% across all three datasets.
> The high concentration of FMPose could also be observed in the comparisons to DiffPose in Fig. 5.
>
> 2. Why straight probability paths are more efficient
>
> We modify the statement in the last line of Section 2.2 as follows: "Based on the experimental results from this study, the straight OT probability path of FMPose demonstrates the better performance, in both prediction accuracy and model complexity, for the task of 2D-3D keypoint lifting compared to the stochastic probability path of DiffPose."
>
> 3. Details about 2D conditioning of DiffPose
>
> For each joint from the estimated heatmap from HRNet, the random sampling strategy draws a set of $k$ candidates based on the corresponding confident scores. The output tensor has the same shape as top-$k$ strategy and can be easily integrated into FMPose. The random strategy has a significantly higher chance to draw bad 2D candidates with low confidence scores, leading to bad generated 3D hypotheses. Adopting the top-$k$ ensures sufficient uncertainty coverage by only considering the $k$ 2D candidates with highest confidence scores (48 for FMPose), and not risking bad 2D candidates to negatively affect the hypothesis set.
>
> 4. Limited metrics on the MPI-INF-3DHP dataset
>
> We followed the evaluation metrics suggested by previous studies for comparable results (DiffPose, Wehrbein et al.). For the MPI-INF-3DHP dataset, it is common in the field to report only the PCK metric as it is the lead metric for handling the evaluation across settings (indoor, outdoor, green-screen) proposed in the original paper from [Mehta et al].
>
> [Mehta et al.], Monocular 3D Human Pose Estimation In The Wild Using Improved CNN Supervision, 3DV, 2017.
>
> 5. FMPose's speed with a high number of steps
>
> As demonstrated in Section 3.3, Equation 6, the selected solver RK2 requires an evaluation at a pseudo-step $\Delta t/2$ to compute the full-step offset. That means the model $f_\theta$ is passed once at the pseudo-step $\Delta t/2$ and another time at $\Delta t$, creating a slight computational overhead. However, it is shown in Table 8 that FMPose with fewer steps already has better performance than DiffPose. The final selection of the number of steps to be 25 is to examine the maximized model performance in term of accuracy in this study.
> In general, the number of steps can be easily altered depending on the downstream tasks.
>
> **Minor comments**
>
> 1. Consider explicitly stating that training is end-to-end
>
> We now added this information to Section 3.4.
>
> 2. Wrong shape of input tensor
>
> Thanks, we fixed to $\mathbb{R}^{J \times 2k}$.
>
> 3. The definition of P-MPJPE
>
> P-MPJPE requires the Procrustes alignment between the predicted and ground-truth 3D poses before computing MPJPE
> Procrustes alignment translates, rotates and scales the predicted 3D pose to the ground-truth pose, eliminating the global orientation and scaling factor from the measurement, and only focuses on the estimated shape and pose (Ionescu et al. 2014).
>
> 4. It is unclear whether the results correspond to top-(k) conditioning or random sampling
>
> In the final sentence of Section 4.3.1, we state that we select top-$k$ version of FMPose for further evaluation, including Table 3 and 4.

---

### Review · Reviewer_tvUM · 2026-04-20

**Summary Of Contributions:**

The main contributions of this submission can be summarized in three points. First, the paper introduces continuous normalizing flows trained with flow matching and optimal transport for probabilistic monocular 3D human pose estimation. Second, it constructs a 2D-3D lifting condition by extracting top-k arguments from 2D heatmaps and processing them with a GCN equipped with learnable connections between human joints. Third, it conducts experiments on the standard Human3.6M protocol, the ambiguous subset of Human3.6M, MPI-INF-3DHP, and 3DPW, showing improved quantitative results over comparable baselines particularly on Human3.6M and MPI-INF-3DHP. The paper also provides ablation studies on the conditioning design, the GCN, the ODE solver, the number of ODE solving steps, and computational complexity.

The main strengths are: (1) the paper reports improvements on both the standard Human3.6M setting and the ambiguous subset; (2) the ablation studies are relatively thorough, covering the GCN-based conditioning, the learnable adjacency matrix, the ODE solver, and the number of solving steps; and (3) the paper includes a complexity comparison with DiffPose and provides some insight into the trade-off between speed, parameter count, and accuracy.

The main weaknesses are: (1) it is not sufficiently disentangled whether the improvement over DiffPose comes from flow matching itself or from the GCN/top-k conditioning design; (2) the central claim of being "faster and more accurate" is not fully consistent with the final-model comparison in Table 8; (3) the 3DPW comparison is explicitly described by the authors as "only for reference," so strong claims on that benchmark should be toned down; and (4) several implementation details relevant to reproducibility are missing or unclear, including at least the specific value of k, the dropout configuration, and the dimensionalities of the GCN and condition vector.

**Additional Comments:**

Overall, the paper has a good core idea. The results on Human3.6M and MPI-INF-3DHP, the evaluation on the ambiguous subset, and the ablations on the conditioning module and solver are strong. However, in the current version, the separation of "what actually caused the improvement" is not sufficiently clear, and the strength of the claims in the Abstract and Conclusion is somewhat ahead of what the experiments fully support. With better-calibrated claims, stronger reproducibility details, and corrections to the internal inconsistencies, this could become a much more convincing paper.

**Audience:**

Yes

**Audience Explanation:**

This paper should be of interest to at least some members of the TMLR audience because it connects recent progress in generative modeling, namely flow matching and continuous normalizing flows, to a concrete structured prediction task: monocular 3D human pose estimation. In particular, the paper raises useful questions about ODE-based alternatives to diffusion models, evaluation on ambiguous poses, the interaction between conditioning design and the generative model, and the trade-off between solver choice, number of steps, and accuracy. Even if the final claims need to be refined, the improvements on Human3.6M and MPI-INF-3DHP, together with the ablations on the conditioning module and solver, are findings that would be worth knowing for part of the TMLR readership.

**Broader Impact Concerns:**

The paper explicitly states that 3D human pose estimation is important for autonomous driving, robotics, and public safety. While more robust monocular 3D pose estimation may benefit such applications, it also has a dual-use nature because it can improve surveillance and person-tracking capabilities. In addition, the authors themselves state in the limitations that FMPose does not use scene interaction and is vulnerable to failures of the 2D detector. Therefore, a short Broader Impact Statement would be useful, covering potential overconfidence in safety-critical applications, misuse of predictions that ignore occlusion, contact, or human-scene interaction, and possible deployment in surveillance-related settings.

**Claims And Evidence:**

No

**Claims Explanation:**

The central empirical results regarding accuracy improvements are generally convincing. In particular, on Human3.6M under the standard setting with H=200, FMPose_top-k reports 41.7 / 30.6 compared with DiffPose’s 44.2 / 32.1. On the ambiguous subset, FMPose_random reports 58.9 / 46.7 / 96.6 / 194.7 compared with DiffPose’s 66.5 / 48.5 / 94.3 / 194.2. These tables support the claim that the method is promising in terms of pose accuracy. Tables 5–7 also indicate that the conditioning module, the GCN, the solver choice, and the number of solving steps affect performance.

However, several of the claims foregrounded by the paper are not sufficiently supported by the current evidence. For example, the Abstract and Conclusion broadly state that FMPose is "faster and more accurate" than diffusion-based methods. Yet in the final-model comparison in Table 8, FMPose with 25 ODE solving steps takes 36.1 ms, whereas DiffPose takes 17.5 ms. Thus, when comparing the final configurations, FMPose is slower than DiffPose.

**Requested Changes:**

1. In the current Table 8, the final 25-step version of FMPose is slower than DiffPose. Therefore, the "faster and more accurate" claim in the Abstract and Conclusion is overgeneralized. The authors should either clearly separate comparisons between final configurations, equal-accuracy settings, and equal-latency settings, or weaken the claim to state that FMPose is faster and accurate only under some step configurations.

2. Please disentangle the contribution of flow matching itself from the contribution of the conditioning design. In the comparison with DiffPose, not only the generative process but also the 2D-3D lifting condition module is changed. Based on the current results, it is not possible to conclude that the main source of improvement is flow matching itself. If possible, the authors should compare diffusion and flow matching under the same conditioning module, or compare Transformer-based conditioning and GCN/top-k conditioning under the same generator. At minimum, the causal wording in the paper should be weakened.

3. Please align the claims about uncertainty with the evaluation. Best-of-H MPJPE-style metrics measure the accuracy of the best hypothesis, but they do not directly evaluate the quality or calibration of the predicted probability distribution. PCK and CPS are useful auxiliary metrics, but if the paper foregrounds uncertainty measurement, it should either add more direct uncertainty or calibration evaluations, or make the uncertainty-related claims more modest.

4. Please clarify and moderate the claims regarding 3DPW. The paper itself states that Table 4 is "only for reference" because the baselines are mesh-based methods and are not directly comparable with the joint-based FMPose. Given this caveat, the Abstract’s claim of large improvements across three benchmarks is too strong. For 3DPW, the authors should either add directly comparable baselines or restrict the claim.

5. Please clarify implementation details necessary for reproducibility. At minimum, the paper should explicitly state the value of top-k, the location and rate of dropout, the dimensionalities used in the GCN, the dimensionality of the condition vector, and when random sampling versus top-k selection is used during training and inference. The current PDF does not provide enough detail for reliable reimplementation.

6. Please correct internal inconsistencies and numerical or unit errors in the paper. Examples include the inconsistency between $z \in R^{J\times 2k}$ and $R^{J\times k}$, the phrase "five random seeds (0–5)," the calculation of the PCK difference based on Table 2, and the "7.6 mm" unit under Table 8. These may look minor, but they matter for readers who evaluate the paper’s precision.

---

> ### Author Response · Authors · 2026-04-30
> **Response to Reviewer tvUM**
>
> We thank the reviewer for their constructive feedback and recommendations. We provide our response to the reviewer's concerns in the following and we have updated the manuscript correspondingly.
>
> 1. Claims about FMPose's performance in the Abstract and Conclusion
>
> We additionally found the equal-accuracy configuration at 7 solving steps, with an equal P-MPJPE at $48.5$mm.
> FMPose with 7 steps has 10.5ms of processing time, $40$\% faster than DiffPose.
> Moreover, as we mentioned in Sec. 4.5, The FMPose with 10 steps is both $16.0$\% faster and more accurate (in both MPJPE and P-MPJPE) than DiffPose.
> Please refer to Tab. 8, steps 7 and 10.
> We now changed the statement in the Abstract to: *"While trade-offs between processing time and precision exist, already in the  equal-accuracy comparison, FMPose exhibits significantly faster processing time than the diffusion model, and also offers another faster and more accurate configuration."*
>
> 2. Verify the contribution of flow matching
>
> Based on the reviewer's suggestion, we included one additional ablation study to Tab. 5 on Transformer-based conditioning, replacing the GCN with the transformer architecture while maintaining the same number of learning parameters.
> Our GCN conditioning has better accuracy ($3$\% on the testset and $2$\% on the ambiguous testset) than the Transformer-based conditioning.
>
> Compared to DiffPose, FMPose with Transformer has $2.7$\% and $7.8$\% improvements on the respective testsets, while FMPose with GCN has $5.6$\% and $9.8$\%, respectively. This demonstrates that the performance contribution of the OT flow matching is roughly the same as the GCN on the testset and significantly larger on the ambiguous testset.
>
> 3. Uncertainty evaluation
>
> We added a new section D, Fig. 5 and Tab. 10 in the Appendix to demonstrate the uncertainty measurement.
> We measure the standard deviations from the $200$ generated 3D hypotheses on three in-the-wild datasets MPI-INF-3DHP, KTH Football, and Leeds Sport Pose.
> Both qualitative and quantitative results show that FMPose produces more concentrated hypotheses compared to DiffPose.
>
> 4. The claims about performance on 3DPW
>
> We now changed the claim about superior performance only on the two datasets Human3.6M, MPI-INF-3DHP in the Abstract, and include an additional statement on 3DPW : *"Additionally, FMPose shows competitive performance on the more challenging 3DPW dataset."*
>
> 5. Additional implementation details
>
> The selection of top-k is shown in Appendix A and Sec 3.4, with a value k of 48 used in all main experiments.
> The dimension of the condition vector is 64.
> Dropout is applied after the first linear layer in the GCN module, with the rate of 0.01.
> Top-k or random sampling is used after the HRNet's prediction.
> All of these information has now been included in the manuscript.
>
> 6. Inconsistencies and errors
>
> We thank the reviewer for pointing this out.
> We have now fixed the inconsistencies and errors.

---

### Decision · Action_Editor_Umxx · 2026-05-14

**Recommendation:** Accept with minor revision

**Additional Comments:**

The paper requires some minor edits to improve consistency in the references. For example, some venue names are abbreviated while others are written in full, and some entries include DOIs while others do not (remove). Additionally, a few grammatical issues and typos remain throughout the paper and should be corrected.

**Audience:**

Yes

**Audience Explanation:**

The paper will be of interest to researchers in the cross-section of ML (diffusion) and 3D human pose estimation.

**Claims And Evidence:**

Yes

**Claims Explanation:**

The paper’s claims are generally supported by convincing experiments, comparisons across multiple benchmarks, and ablation studies.